# STUDENTEVAL:
# A BENCHMARK OF STUDENT-WRITTEN PROMPTS FOR LARGE LANGUAGE MODELS OF CODE

## ABSTRACT

Code LLMs are being rapidly deployed and there is evidence that they can make professional programmers more productive. Current benchmarks for code generation measure whether models generate correct programs given an expert prompt. In this paper, we present a new benchmark containing multiple prompts per problem, written by a specific population of non-expert prompters: beginning programmers. STUDENTEVAL contains 1,749 prompts for 48 problems, written by 80 students who have only completed one semester of Python programming. Our students wrote these prompts while working interactively with a Code LLM, and we observed very mixed success rates. We use STUDENTEVAL to evaluate 12 Code LLMs and find that STUDENTEVAL is a better discriminator of model performance than existing benchmarks. We analyze the prompts and find significant variation in students' prompting techniques. We also find that nondeterministic LLM sampling can mislead students into thinking that their prompts are more (or less) effective than they actually are, which has implications for how to teach with Code LLMs.

## 1 INTRODUCTION

Large language models of code (Code LLMs) power coding assistants that are rapidly reshaping how programmers write code. Researchers have studied their impact on programmer productivity (Vaithilingam et al., 2022; Ziegler et al., 2022; Barke et al., 2022), identified real concerns about potential harms (Dakhel et al., 2022; Mozannar et al., 2022; Sandoval et al., 2023; Pearce et al., 2021; Aghakhani et al., 2023), and considered how they could help students learn (Leinonen et al., 2022; Finnie-Ansley et al., 2022; Jayagopal et al., 2022). Fundamental to these studies, and to tool adoption, is the assurance that the underlying models work effectively and consistently.

Code LLMs are commonly evaluated using benchmark suites that cover a wide variety of problems. Popular benchmarks such as HumanEval Chen et al. (2021) and MBPP Austin et al. (2021) consist of many problems from varying areas of computing, accompanied by a single expert-written prompt. Achieving good performance on these benchmarks indicates that a model will perform well across many programming tasks, *assuming that the user can write prompts equally as well as the expert*.

In this paper, we present a Code LLM benchmark in a context where this assumption does not hold: beginning programmers using Code LLMs. Our STUDENTEVAL dataset contains 1,749 student-written prompts (with expert-written test cases) which we use to benchmark several Code LLMs. STUDENTEVAL is constructed using a novel approach that sets it apart from prior work in three key ways. **1)** Existing benchmarks (Chen et al., 2021; Hendrycks et al., 2021; Austin et al., 2021) have prompts authored by more experienced programmers, whereas STUDENTEVAL has *prompts authored by students who have only completed one computer science course*. **2)** Existing benchmarks contain tricky problems designed to stress-test the problem solving capabilities of Code LLMs. In contrast, STUDENTEVAL has problems that are *easily solved with expert-written descriptions, but often fail with student-written descriptions*. **3)** Existing benchmarks only have a single prompt per problem, whereas STUDENTEVAL *has on average 36 prompts per problem, representing a variety of prompting skill levels*. This diversity provides a way to explore what it means to write a "good" prompt and to measure the impact of prompt wording choices.

The STUDENTEVAL problems target a specific skill level and provide a diverse set of prompts for each problem along with expert-written test cases. Students wrote English descriptions of these problems in an iterative manner in collaboration with a Codex model. Each of the 48 problems in STUDENTEVAL contains at least 14 different prompts. Notably, these prompts exhibit the variations in technical vocabulary and lack of familiarity with how to describe code that are common with beginning students. While other researchers have considered novice student interactions with Code LLMs (Leinonen et al., 2023), STUDENTEVAL is the first benchmark based on student interactions. This framing provides significant insight into Code LLM reasoning capabilities outside of the educational context.

Our key contributions are:

- We present STUDENTEVAL, a benchmark consisting of 1,749 student-written descriptions of 48 programming problems.

- We identify four key subsets of the STUDENTEVAL benchmark, consisting of descriptions that pass (fail) on the first (last) attempt by a student, and evaluate these subsets on 12 state-of-the-art Code LLMs. Our results show that STUDENTEVAL is better able to discriminate between models than the popular HumanEval benchmark.

- We conduct an in-depth analysis of the prompts and find that even successful student prompts lead models to generate multiple semantically distinct programs.

## 2 BACKGROUND

Existing code generation benchmarks pair natural language descriptions of code with test cases to check the validity of generated programs. The two most commonly used benchmarks, HumanEval (Chen et al., 2021) and MBPP (Austin et al., 2021), are in Python. There are also multi-language benchmarks that translate problems from one language to another (Athiwaratkun et al., 2023; Cassano et al., 2023). Finally, there are alternate benchmark formats, including multi-turn evaluation (Nijkamp et al., 2022) and docstring generation (Lu et al., 2021).

**General-purpose benchmarks**   Most benchmarks have a single natural language description per problem, which is typically written by an expert. There are exceptions that scrape the web or crowdsource (Hendrycks et al., 2021; Lai et al., 2022; Amini et al., 2019), but the dominant trend is for experts to generate benchmarks themselves. Expert-written prompts can provide wide coverage, but come with limitations. First, they have a *single* prompt per problem. Consider this HumanEval (Chen et al., 2021) prompt:

*Imagine a road that's a perfectly straight infinitely long line. n cars are driving left to right; simultaneously, a different set of n cars are driving right to left. The two sets of cars start out being very far from each other. All cars move in the same speed. Two cars are said to collide when a car that's moving left to right hits a car that's moving right to left. However, the cars are infinitely sturdy and strong; as a result, they continue moving in their trajectory as if they did not collide. This function outputs the number of such collisions.*

While the correct solution is simply $n^2$, the prompt is designed to be purposefully confusing. This means that models succeed or fail based on this *specific phrasing*. Having a single prompt precludes explorations of how crucial specific word choice, grammar, etc. is to model success. STUDENTEVAL's non-expert construction allows us to better analyze what makes a successful prompt, as we have at least 14 prompts per problem, and it helps differentiate variations in model development that contribute to success.

Second, existing benchmarks contain problems at widely varying difficulty levels. Compare the prompt above, which requires mathematical reasoning that might challenge many programmers, with a trivial problem from the same benchmark (Chen et al., 2021): *Return length of given string*. Although these benchmarks succeed in capturing a wide range of programming tasks, it is difficult to interpret their results as evidence that a model will or will not serve the needs of a particular group of programmers, since their results aggregate over problems at very different skill levels.

**Domain-specific benchmarks**   There are also a number of domain-specific benchmarks which, rather than present a range of tasks, focus on a more narrow domain. Two notable such benchmarks

| Function signature (visible) | `def convert(lst):` | |
|---|---|---|
| **Reference implementation** (hidden) | `return ''.join([ chr(i+65) if i >= 0 else " "`
`    for i in lst ]).split()` | |
| **Expert tests** (visible to student; hidden from model; automatically run on generated code) | **Input** | **Expected Output** |
| | `[0,1,2,3]` | `['ABCD']` |
| | `[0,-1,1,-1,2]` | `['A','B','C']` |
| | `[1,1,1,-1,25,25,-1,0,1,2]` | `['BBB','ZZ','ABC']` |
| **Student description** (pass@1 = 0.8) | takes a list of numbers. Create a ABC list with the capital letters in the alphabet and create an answer string. Iterate through the input list, if there is "-1" then add ' ' to the answer string, or otherwise, add the letter with the corresponding index of the answer string. Split the answer string at ' '. return the answer string. | |
| **Student description** (pass@1 = 0.0) | Assign a number from 0~25 to each alphabet, and create a list of string of alphabetical letters based on their assigned numbers in the lst. When there is -1 in the lst, create a new string and add it to the list. Return a list of created strings. | |

Figure 1: An example problem from STUDENTEVAL. Our web-based experiment platform shows students the signature and expert-written tests. When students submit their description, we use a Code LLM to generate code, test it, and flag any failed tests for the students. STUDENTEVAL has multiple student-written descriptions for each problem.

are DS-1000 (Lai et al., 2022) and MathQA-Python (Austin et al., 2021). Like domain-specific benchmarks, our benchmark targets a specific population of programmers; however, we target a particular skill level rather than a specific application area. We also provide numerous prompts per problem from our non-expert annotators.

## 3 THE STUDENTEVAL DATASET

In this section we describe STUDENTEVAL, a many-prompt-per-problem benchmark that targets a specific programmer skill level. The dataset consists of 1,749 English-language prompts for 48 programming problems, with at least 14 prompts per problem. All prompts were written by university students who had completed 1 semester of computer science in Python (CS1), but no subsequent CS courses. These students represent a population of programmers with a uniform knowledge base, which allows us to select problems that should be solvable for all participants.

**Problem Selection and Format** We compiled a suite of 48 programs that closely resembled the kinds of problems that are familiar to students. The problems exercise a variety of Python features. The majority of problems were pulled directly from CS1 course materials, with light modifications to avoid publishing answers to assignments still in use. Thus, all participants should be able to understand and solve the problems by directly writing solutions in Python; we explore whether they are also able to describe them in natural language so that code generation models can solve them. For topic diversity, we came up with 8 core concepts that were covered by all institutions and selected 6 problems from each concept category. The categories are lists, loops, strings, conditionals, math, nested data, sorting, and dictionaries.

Each STUDENTEVAL problem consists of three components: a function signature, a reference implementation, and 3+ test cases that achieve high coverage on the reference implementation (Figure 1). When we gather student data, which we describe below, we show participants only a function's signature and test cases. From this information, they produce a description, which we we automatically validate using the problem's test cases.

| Subset | Items | Word Count |
|--------|-------|------------|
| First Failure | 450 | 28.8 (25.5) ± 16.7 |
| First Success | 187 | 28.8 (25.0) ± 17.4 |
| Last Failure | 205 | 35.9 (30.0) ± 22.6 |
| Last Success | 185 | 37.8 (35.0) ± 18.4 |

(a) Sizes and word counts.

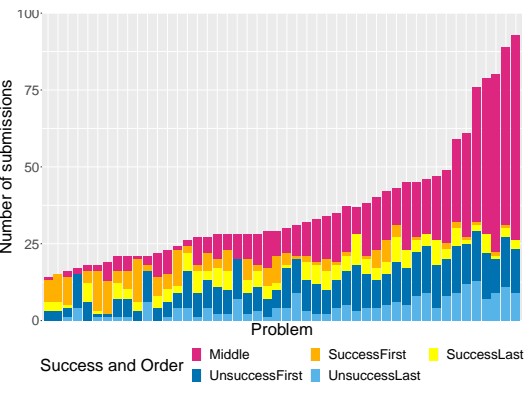

(b) Attempts per problem.

Figure 2: The four subsets of STUDENTEVAL.

**Problem Validation**   We validated our problems in several ways. For common problems (e.g. factorial), an LLM can produce a working implementations from the function name alone. To weed out these problems, we produced Codex generations from each function signature with no docstring and measured mean pass@1 rate. Overall, the mean pass@1 for our signatures without docstrings is 0.0519 with a variance of 0.0364. The maximum pass@1 is 0.925, for the problem `exp`.

We also validated the test suites associated with each problem. The test cases serve two roles in our dataset collection: they help students understand the problem, and they ensure that the LLM-generated solutions are correct. Liu et al. (2023) give evidence that the test cases that accompany widely-used Code LLM benchmarks frequently miss important corner cases. To avoid this pitfall, we use both test coverage and mutation testing of the expert-written solution to ensure that the test cases in STUDENTEVAL are adequate. Unlike the tests in Liu et al. (2023), the STUDENTEVAL tests need to be simple enough to be understandable by students who have only completed CS1. Therefore, we strive to strike a balance between exhaustiveness and comprehensibility. Every problem has 3–4 tests that achieve 100% code coverage. In fact, we ensure that every problem has three tests, even if they are not necessary to achieve coverage, in order to aid participant understanding of the problems. Mutation testing (Jia & Harman, 2010) is a more rigorous way than coverage to measure the quality of a test suite, and we used MutPy (Hałas, 2013) to compute mutation scores. All mutation scores below 90 are either the result of MutPy generating no mutations at all, or generating a technical correct mutation that still passes tests.

**Gathering 1,749 Student-Written Prompts**   We recruited 80 beginning CS students from three U.S. higher education institutions to build the STUDENTEVAL benchmark. The study was IRB-approved - we obtained both consent prior to and verbal assent during the study. We conducted the study over Zoom, using a web-based application designed specifically for STUDENTEVAL (see Appendix). This application presents the function signature and tests for one problem at a time. Students enter a problem description into a text box. After they submit, our server constructs a prompt with the function signature and their problem description formatted as a Python docstring, and sends this prompt to Codex to produce the function body. The server then tests the function in a sandbox and presents the test results to the participant. Students had the option to reattempt the problem or move on to the next problem. Participants completed three tutorial and 8 STUDENTEVAL problems in ≈75 minutes, receiving a $50 gift card for participation.

**Dataset Subsets and Basic Statistics**   Students generated 1,749 prompts in total, with an average of 36 prompts per problem. There are significant variations in how the prompts differ from each other: many are small, iterative changes (+/- a few words) whereas a student's first, last, and successful prompts tend to vary significantly from others. To refine the dataset for evaluation, we break the STUDENTEVAL dataset into four disjoint subsets (Figure 2a): students most frequently failed to solve problems on their first attempt, and this is the largest subset of problems (*First Failure*); about half as many first attempts were successful (*First Success*); slightly fewer students gave up after multiple attempts (*Last Failure*); and others succeeded after multiple attempts (*Last Success*). (These subsets

Table 1: Mean pass@1 for the models that we evaluate on the four subsets of STUDENTEVAL.

| Model (Size) | First Failure | Last Failure | First Success | Last Success | HumanEval |
|---|---|---|---|---|---|
| GPT-3.5-Turbo-0301 (?) | 10.86 | **12.41** | 44.84 | 47.40 | 48.1 |
| Phi-1 (1.3B) | 11.28 | 8.37 | 59.16 | 36.36 | 51.22 |
| StarCoderBase-1B (1B) | 1.77 | 1.21 | 24.86 | 13.00 | 15.17 |
| StarCoderBase-3B (3B) | 5.91 | 5.66 | 51.73 | 32.20 | 21.46 |
| StarCoderBase-7B (7B) | 5.49 | 6.82 | 62.35 | 46.42 | 28.37 |
| StarCoderBase (15.5B) | 7.82 | 6.74 | 65.28 | 51.74 | 30.40 |
| Code-Llama-Py-7B (7B) | 6.51 | 8.59 | 66.88 | 55.36 | 40.48 |
| Code-Llama-Py-13B (13B) | 9.56 | 9.33 | 70.22 | 62.26 | 42.89 |
| Code-Llama-Py-34B (34B) | **11.40** | 10.14 | **73.51** | **64.65** | **53.29** |

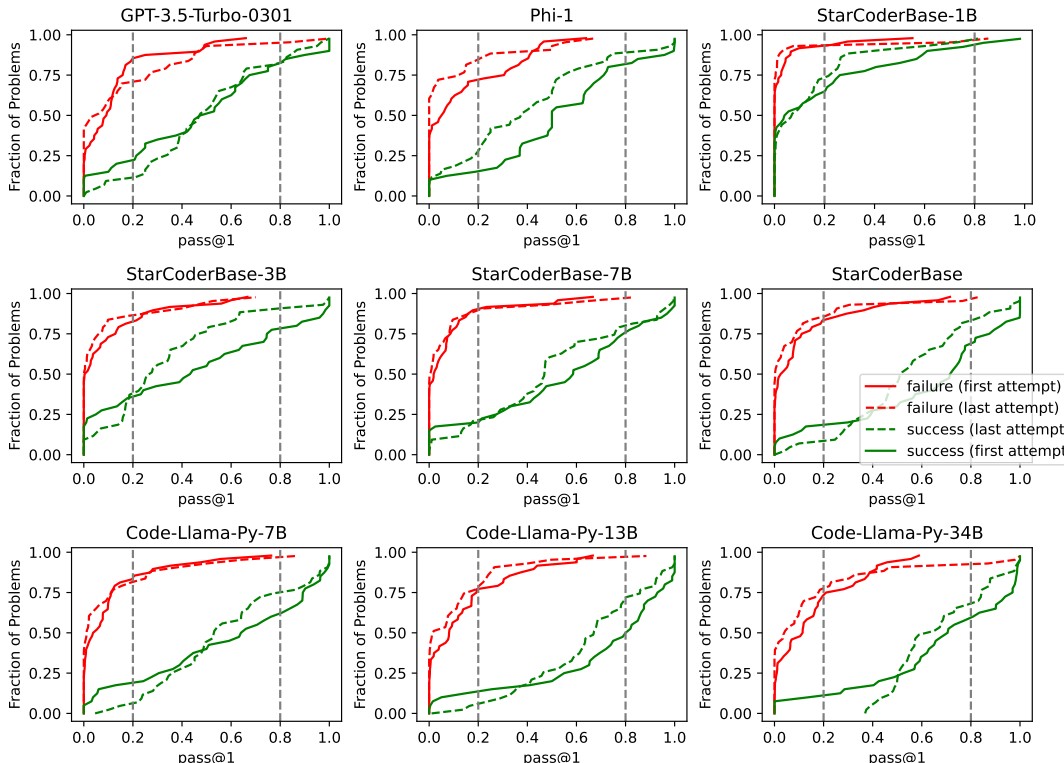

Figure 3: CDFs of mean, per-problem pass@1 for several Code LLMs on the four subsets of STUDENTEVAL. The $y$-axis reports the fraction of problems in each subset and the $x$-axis reports the mean pass@1 for all student-written descriptions for those problems.

omit the "Middle" prompts (Figure 2b) that were all failures and neither a first nor last attempt.) Figure 2a shows that the *Last* descriptions are significantly longer than the *First*, which suggests that students keep adding detail, even when starting afresh may be the better approach.

## 4 RESULTS

We evaluate 12 Code LLMs. This section focuses on gpt-3.5-turbo, the three "Python specialist" Code Llama models (Baptiste Rozière et al., 2023), the four StarCoderBase models (Li et al., 2023), and Phi-1 (Gunasekar et al., 2023). Appendix E presents results for several other models. We confirm that none of the STUDENTEVAL prompts appear in The Stack, which is the open training dataset for StarCoderBase and other models.

As with other benchmarks, we use hidden unit tests to evaluate the correctness of model-generated code. To account for their nondeterministic output (e.g., when using a sampler during inference), we adopt the now standard *pass@1* metric (Chen et al., 2021). Pass@1 is an estimate of the probability that a Code LLM will produce a correct solution for a prompt that passes all hidden unit tests in one shot. We generate 200 samples for each prompt to calculate pass@1, which is also standard.

## 4.1 HOW DO MODELS PERFORM ON STUDENTEVAL?

Table 1 reports the mean pass@1 rate for every model on the four subsets of STUDENTEVAL. We also include HumanEval pass@1 rates for comparison.

**Code Llama models perform best on STUDENTEVAL**    We find that *the Code Llama models significantly outperform all other models on the* First/Last Success *prompts.* The 13B model outperforms StarCoderBase-15B, the closest competing model, by 5-10% (absolute). The 34B model performs even better.

**STUDENTEVAL exposes a bigger gap between larger and smaller models than HumanEval** HumanEval is the de facto standard coding benchmark: Code LLM highlight their HumanEval scores, and there are several LLMs that strive to do well on HumanEval while remaining relatively small. However, we observe that the difference between pass@1 rates for the larger and smaller models is more substantial with STUDENTEVAL than HumanEval. 1) For the StarCoderBase models, pass@1 on *Last Success* is almost 4x higher with the 15B model vs the 1B model, but the gap is much smaller (2x) on HumanEval. 2) Phi-1 (1.3B) approaches the performance of Code Llama (34B) on HumanEval. However, Code Llama is 1.7x better on *Last Success* than Phi-1. Phi-1's HumanEval performance comes from its "textbook quality" training data. Unfortunately, students don't write textbook quality prompts, so we conclude that natural data is better textbook-quality data for a model to follow student prompts.

## 4.2 VARIATION IN PASS@1

Most Code LLM papers only report mean pass@1 for a benchmark, averaging over problems with widely varying pass rates. Because STUDENTEVAL contains multiple prompts per problem, it illuminates the extent to which luck plays a role in whether a Code LLM produces the right answer for a user. In Figure 3, we group all prompts by problem, so the plots show the percentage of problems ($Y$) with pass@1 lower than the indicated value ($X$).

For a given model, let us define a *reliable failure* to be a prompt that is in *First/Last Failure* but has pass@1 greater than 0.8 (problems to the right of the dashed line at 0.8 in the CDF). These are cases of bad luck: the prompt failed when the student tried it, but turns out to be reliable with the given model. We find that GPT-3.5-Turbo-0301 and StarCoderBase have one and two reliable failures each. Similarly, let us define an *unreliable success* as a prompt that is in *First/Last Success* but has pass@1 lower than 0.2. These are cases of good luck: the prompt worked once for a student, but that success is hard to reproduce. We find that nearly 10% of successful prompts are unreliable for smaller models but less than 3% are unreliable with the larger models. (Table 2 in the Appendix has a table of counts.)

Overall, we believe these results have implications for model selection. It is not adequate to optimize a model to achieve high pass@1 on any benchmark, including STUDENTEVAL. An ideal Code LLM would maximize pass@1 and minimize its variability.

## 4.3 PARTICIPANT SUCCESS RATES

Examining prompt success rates by participant shows that our dataset represents a wide spectrum of prompting ability levels (Figure 4). Although some participants achieve prompt success rates over 50% with StarCoderBase, a large number struggle to write reliably successful prompts.

A participant might have a low success rate for various reasons. They might not be very skilled at writing prompts, describing the problem to be solved vaguely or even incorrectly. Or they may be writing clear explanations of the problems, but in a style that the model does not understand. Thus, a

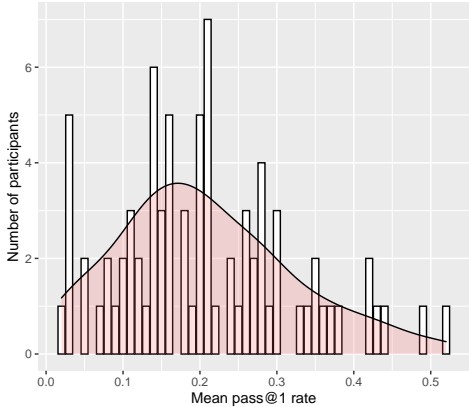

Figure 4: Participant mean pass@1 rates with StarCoderBase

Figure 5: TF-IDF values for overlapping words in the top 25 words for all subsets.

low success rate does not necessarily indicate a lack of skill on the part of the participant; it can also indicate that models systematically struggle with particular ways to describe code.

## 5 What Makes a Successful Prompt?

### 5.1 Trends in Student Word Choice

To explore the relative importance of different words, we tokenized the STUDENTEVAL prompts and computed TF-IDF values for the four data subsets and then calculated the mean score per word across all prompts. Figure 5 shows the mean frequency matrix for words that appear in the top 25 for *all* subsets and Figure 11 (Appendix) shows the mean scores for the top 25 words individually.

The top words are a mix of English and Python terms, including many related to types, sequencing, or choice. return/s as part of Figure 5 confirms an anecdotal observation: Codex defaults to printing output, which is not conducive to our test-driven evaluation, so specifying "return" may be a learned behavior. We observe a similar trend for specifying parameter names. The lack of large differences between scores and across subsets may be due to the data size or average prompt length (Figure 2a).

### 5.2 Statistical Significance of Prompt Wording

We fitted mixed-effects regression models to the data to test the impact of prompt length and wording choices. All models include random effects for problems and use StarCoderBase pass@1 rates as the response variable. For vocabulary-level features, we use indicator variables: 1 if the prompt uses the word and 0 otherwise. *Appendix D.3 provides full estimate tables from mixed-effects models; we highlight key findings below.*

**Length** Contrary to our expectations, we observed a statistically significant positive effect of prompt length on pass@1 rates ($p$=0.007). However, this finding seems driven by last submissions, where successful prompts are on average longer; the average length is similar for passing and failing first prompts (Figure 2a). Qualitatively, we have observed that students tend to add more detail on subsequent attempts rather than modifying their earlier text, which likely contributes to this finding.

**Input/output word choice** We found a significant positive effect of mentioning "return" in the prompt ($p$<0.0001). This likely resolves the problematic ambiguity associated with prompts that mention "output" rather than specifying whether the function should return or print (Figure 6b).

**Datatype mentions** We explored the effect of mentioning dictionaries, lists, and number types, as well as including instances of lists and dictionaries in the prompt. We found a reliable positive effect

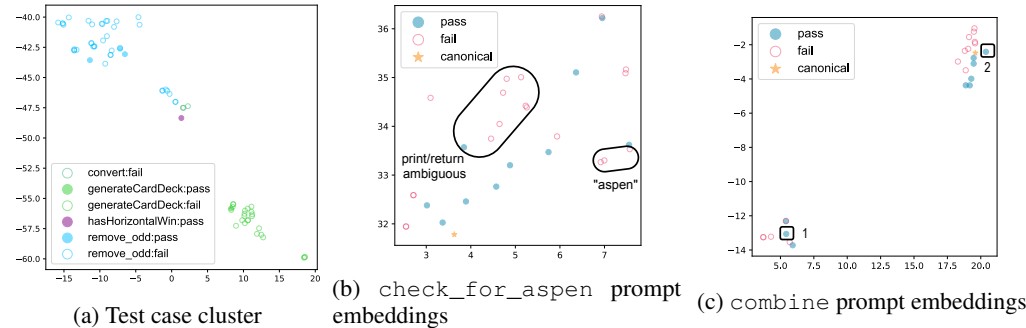

(a) Test case cluster

(b) `check_for_aspen` prompt embeddings

(c) `combine` prompt embeddings

Figure 6: Prompt embeddings generated using StarCoderBase and reduced using t-SNE.

of mentioning "list" ($p$=0.02), and a borderline negative effect of mentioning "array" ($p$=0.053). This suggests that StarCoderBase is sensitive to Python terminology conventions.

**Function and parameter names** We found no reliable effect of mentioning the parameter names in the prompt, but a significant negative effect of mentioning the function name ($p$=0.02).

## 5.3 INSPECTING VISUAL REPRESENTATIONS

We generated embeddings of each prompt from the last-layer attention weights of the StarCoderBase model in order to explore prompt similarities and differences. Figure 6 shows some key clusters of embeddings plotted using t-SNE (Van der Maaten & Hinton, 2008); Appendix D.4 contains a plot of all clusters.

**Multiple prompt formulations exist.** The prompts for some problems form multiple clusters, indicating multiple ways to describe the task. The prompts for the `combine` problem form two clusters (Figure 6c). The top right cluster contains several succinct prompts, as exemplified by Prompt 2: *Combine lists from l1 to lists from l2.* The bottom left prompts do more "handholding", providing detailed step-by-step directions. For instance, Prompt 1 spells out multiple steps: *Takes an input of two lists, l1 and l2, each of which also contains lists. It combines the first list in l1 with the first one in l2, then continues for all items in l1 and l2. It outputs this final list which is a combination of l1 and l2.* Both approaches can generate passing programs; future work could explore whether there are style differences between the programs generated by different prompting methods.

**Errors and ambiguities pattern together** Examining problem sub-clusters also reveals patterns in prompting failures. In Figure 6b, for instance, there is a sub-cluster of prompts that are ambiguous about whether the function should print or return the desired value. Although a human might be able to disambiguate, these are unreliable prompts: the model may sometimes generate a solution using `print` and sometimes using `return`. Another sub-cluster consists of prompts that contain the string "aspen" (lower-case) rather than "Aspen" (upper-case), causing the generated code to fail test cases.

**Certain prompting styles are challenging** Although most prompt embeddings cluster by problem, a handful of clusters contain prompts for multiple problems, representing cases where the model struggles to distinguish among problem descriptions. One prompting style that students use is to describe the function's behavior in terms of expected input/output pairs. For instance, *If the number is below 10, make it 10 [...]* is a prompt that uses this strategy for `increaseScore`.

Although there are passing examples of this style, it does not seem to work well for problems that involve more complex data, such as nested lists or dictionaries. Figure 6a shows a cluster of prompts that give examples of lists that the function should return. These prompts describe different problems, yet their embeddings cluster together away from the clusters of their respective problems, indicating that the model may struggle to differentiate these rarer values. This style of prompt is likely to be well-understood by humans, yet works poorly for current code generation models.

| Subset | #Functions |
|--------|-----------|
| failure (first attempt) | 2.2 (2.0) ± 1.6 |
| failure (last attempt) | 2.4 (2.0) ± 1.6 |
| success (first attempt) | 1.9 (1.0) ± 1.3 |
| success (last attempt) | 2.2 (2.0) ± 1.3 |

(a) Mean (median) & stddev. of the number of functions produced StarCoderBase for each prompt.

The function takes a string of text as an input. For words in the string with an odd number of letters, every other letter is capitalized starting with the first letter. For words in the string with an even number of letters, every other letter is capitalized starting with the second letter.

(b) A *First Success* prompt that produces 7 different functions.

Figure 7: StudentEval prompts can be ambiguous to LLMs and produce several distinct functions.

### 5.4 Ambiguity in Prompts

The previous sections, and prior work on Code LLMs, ask if models produce correct code for a given prompt. However, it is also possible to have an ambiguous prompt that generates several *semantically* different functions. Testing semantic equivalence of Python functions is of course undecidable. But, we compute a lower bound on the number of semantically different functions generated by a prompt as follows. For each completion of a prompt, we use the inputs from the expert-written test cases as a vector of examples. We run each completion on each input and collect a vector of outputs that form the *test signature* of the function Udupa et al. (2013). When two functions have distinct test signatures, that is proof that they are semantically different. Identical results are inconclusive.

Figure 7a summarizes the result of this experiment on each subset of STUDENTEVAL. As expected, prompts in the Success subsets are more reliable: they generate fewer functions on average than prompts in the First/Last Failure subsets. What is more surprising is just how many different functions a single prompt can generate. Even prompts that are relatively clear to human readers, such as the one in Figure 7b, can generate many different functions. Inputting the prompt shown in Figure 7b to StarCoderBase generates completions that contain at least *seven* semantically distinct functions.

This highlights the importance of evaluating prompt *reliability*. Although the prompt shown in Figure 7b happened to produce a passing prompt during the experiment, in reality, this was partly due to luck; the participant was fairly likely to see a failure on their first submission attempt.

This finding has clear implications for the use of code generation models as teaching tools in educational contexts (see discussions in Finnie-Ansley et al. (2022); Leinonen et al. (2023)): reliability issues may both mislead students into thinking their descriptions are clearer than they really are, and mislead them into over-complicating descriptions that are straightforward to a human, but unreliable for prompting code generation models.

## 6 Conclusion

We present STUDENTEVAL, a large benchmark for Code LLMs, where the prompts are written by students who have completed one semester of Python. A key feature of STUDENTEVAL is that it has multiple prompts per problem from repeated attempts and from multiple students. We show that larger models are more capable of following student-written instructions than smaller models. We also find that many student-written are unreliable (have low pass@1): students get lucky (or unlucky) when using Code LLMs. Finally, we investigate several hypotheses of what makes a good prompt.

**Limitations** Students wrote the prompts interactively while using a Codex model. It is likely that they would have revised their problems differently with a different model. STUDENTEVAL only has student-written prompts and is not representative of prompts written by experienced programmers. But, we believe it is valuable to have Code LLM benchmarks that focus on non-experts.

### Ethics Statement

Our work was conducted in accordance with approval from our Institutional Review Board. Potential harms to student participants were a first-class consideration in the design. We sought to address power dynamics and protect participant autonomy with a number of measures. We collected data in

an opt-in manner, outside of the classroom, and with informed consent. The researcher conducting the study was not affiliated with the student participant's institution. Students were asked to complete programming assignments with familiar content and were alerted to potential discomfort caused by interacting with an AI-based tool. All identifying information has been removed from the dataset prior to release.

The main use case for this dataset is to train and evaluate ML-driven technology. Releasing this dataset may lead to the development of technology that we would not build ourselves; for instance, attempts to automate education in a way that would negatively impact the educational experience of students. Future users may generalize results from the dataset beyond what is appropriate; we do not present our dataset as a "standard", but as a (necessarily incomplete) snapshot of early CS education in the USA in the 2020s.

Finally, this research was only possible due to model access and funding. There are ongoing ethical concerns in this research community about access to models and infrastructure. The evaluation of the dataset in this paper centers both open-source and small-scale models, but fully addressing these issues should be a priority for the broader community.

## REPRODUCIBILITY

We plan to release the full dataset via the Open Science Framework. The main data file is included as part of the Appendix. The Appendix also contains a "Datasheet for Dataset" outlining pertinent dataset information. This dataset was collected via a human subjects experiment. We provide pertinent screenshots and text from the tutorial in the Appendix. Consent information will be shared publicly upon acceptance. To obtain the benchmark results from the 12 LLMs, we used at most 2 weeks of GPU time on an H100 GPU.

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
