# OpenReview forum: "StudentEval: A Benchmark of Student-Written Prompts for Large Language Models of Code"
_ICLR.cc/2024/Conference — ICLR 2024 Conference Withdrawn Submission_

### Official Review · Reviewer_Z4ry · 2023-10-28

**Soundness:** 3 good
**Presentation:** 3 good
**Contribution:** 3 good
**Rating:** 6
**Confidence:** 5

**Summary:**

This paper unveils the STUDENTEVAL benchmark, a distinctive tool tailored for gauging the proficiency of Code LLMs using prompts penned by novice programmers. Encompassing 1,749 student-crafted descriptions spanning 48 coding challenges, STUDENTEVAL proves to be a more discerning measure of model performance than its contemporaries. Delving deeper, the authors discern that even the most apt student prompts can guide models towards churning out a diverse range of semantically varied programs. In summation, this paper heralds a new dawn for the LLM community. It not only introduces a benchmark of novel design but also illuminates the avenues through which LLMs can be fine-tuned to adeptly interpret and act upon prompts from budding programmers, especially in the realm of code assistance.

**Strengths:**

1. **Innovative Benchmark Design**: The paper introduces the STUDENTEVAL benchmark, which stands out due to several unique features. Distinctly, it capitalizes on prompts penned by novice programmers, a departure from conventional benchmarks that typically rely on prompts from seasoned professionals. Moreover, the incorporation of multiple prompts for each problem facilitates a more granular evaluation of model efficacy. A key insight unveiled by the authors is that even the most adept student prompts can inadvertently steer models towards producing a spectrum of semantically varied codes — a revelation that is groundbreaking in its own right.

2. **Clarity and Structure**: The document is impeccably articulated, boasting a lucid narrative complemented by logically sequenced sections and sub-sections. There's commendable transparency in detailing the benchmark's design, its evaluative approach, and the models chosen for the assessment. The analysis, particularly of the prompts, is rendered in an approachable fashion, ensuring it is digestible for a broad readership.

3. **Significant Contributions**: This work makes pivotal strides on multiple fronts. Foremost, the STUDENTEVAL benchmark sets a new gold standard for appraising Code LLMs, especially with prompts emanating from fledgling programmers. By offering multiple prompts for each challenge and delving into the intricacies of prompt quality, the paper sheds light on the intricate art and science of crafting potent prompts for Code LLMs. Additionally, the rigorous evaluation of 12 state-of-the-art Code LLMs furnishes a valuable yardstick for comparative model performance.

**Weaknesses:**

1. **Absence of Error Case Analysis**: Though the paper effectively evaluates Code LLMs' performance on the STUDENTEVAL benchmark, it falls short in providing a meticulous dissection of the specific errors manifested by these models. A more comprehensive insight into the precise nature and categories of mistakes these LLMs are prone to would have been invaluable.

2. **Scale Impact on Noisy Prompts**: The paper lacks a systematic exploration of the influence of model scale, especially concerning the noisy nature of student-written prompts. A deeper dive into understanding the types of errors LLMs are susceptible to, and how these errors could potentially be mitigated as the model scales (from 1B to 34B and upwards to ~200B like ChatGPT), would have enriched the analysis.

3. **Cultural Diversity Overlooked**: The authors rightly acknowledge the paper as a "snapshot of early CS education in the USA during the 2020s." However, a potential oversight lies in the lack of discussion surrounding the impact of cultural diversity on prompts and, by extension, on LLMs. Given that students from varying cultural backgrounds, including native English speakers and non-native English speakers, could possess distinct cognitive frameworks, their interaction with LLMs and their prompt construction might differ. In this benchmark dataset, a deeper exploration of how these cultural nuances influence LLM's prompt interpretation would have been a compelling addition to the paper.

**Questions:**

Please see weaknesses. I will update my evaluation after the discussion.

**Details Of Ethics Concerns:**

N.A.

---

### Official Review · Reviewer_8Caj · 2023-10-31

**Soundness:** 3 good
**Presentation:** 3 good
**Contribution:** 2 fair
**Rating:** 5
**Confidence:** 3

**Summary:**

This paper presents a new Code LLM benchmark dataset, containing 1,749 student-written prompts over 58 problems. The authors evaluate different code LLMs on the benchmark, and have several empirical findings.

**Strengths:**

- A evaluation dataset for code LLMs is constructed.
- One key difference of the constructed benchmark is that multiple prompts for one program problem are provided. This leads to quality analysis of prompts, and thus has the value to teach students/developers how to write better prompts.
- Based on the analysis of different prompts, some findings are summarized.

**Weaknesses:**

- The dataset contains only 58 problems. As a comparison, the MBPP benchmark consists of around 1,000 programming problems.

- The authors claim that the STUDENTEVAL is a new benchmark for evaluating code LLMs. But why shall we use low-quality prompts to test LLMs? What makes it necessary? The authors may consider to re-organize the paper in this vein.

- Although the authors list a few findings based on the analysis of the prompts, most of the findings are not very interesting and actionable. It would be better if the authors could summarize the key findings that are beneficial for future prompt writing.

**Questions:**

- The authors claim that the STUDENTEVAL is a new benchmark for evaluating code LLMs. But why shall we use low-quality prompts to test LLMs? What makes it necessary?

---

### Official Review · Reviewer_RtHt · 2023-11-01

**Soundness:** 3 good
**Presentation:** 4 excellent
**Contribution:** 2 fair
**Rating:** 3
**Confidence:** 4

**Summary:**

This paper presents a benchmark for written prompts in the context of CS education who have completed only one Python course. The authors collected 1,749 prompts for 48 problems, written by 80 students.
The authors empirically evaluated prompts with nine code LLMs.

**Strengths:**

1. Dataset paper for benchmarking prompts for CS education
2. Different prompts for the same problem may identify students' weakness

**Weaknesses:**

This paper has limitations from two perspectives: LLM benchmarking and CS education.

From an LLM benchmarking perspective, the authors claimed that the dataset contains an average of 36 prompts per problem (Section 1: Introduction).

However, as presented in Table 1 in the main paper and Table 2 in the Appendix, the prompts are mostly unreliable as shown by pass@1. Thus, evaluating the variation of prompting for a single problem is not reliable.

From a CS education perspective, it is not clear whether the authors assessed i)how students' prompts for the same problem evolved, ii) did the students understood the underlying programming concepts and iii) did students understand why their prompt passed/failed.
An easier approach could have been done by deploying an online survey to assess underlying programming concepts.

**Questions:**

N/A

---

### Official Review · Reviewer_4LKr · 2023-11-01

**Soundness:** 2 fair
**Presentation:** 2 fair
**Contribution:** 2 fair
**Rating:** 3
**Confidence:** 4

**Summary:**

Existing benchmarks to evaluate the performance of Code LLMs are run on professional and well-designed prompts written by experienced programmers. However, it’s challenging for a beginner user to write prompts in an expert fashion, which creates a discrepancy in studying and leveraging the power of Code LLMs. Therefore, the authors proposed a new Code LLM benchmark called StudentEval which is written by beginner programmers. They collected 1749 written prompts for 48 programming problems and investigated the key components toward successful prompts with diverse Code LLMs.

**Strengths:**

* Originality: This benchmark is novel and more practical since most users are non-expert programmers who are not able to write correct and professional prompts to guide LLM to generate high-quality inference outputs. This fills the gap between other existing benchmarks and the usage of normal users.
* Significance: In practical scenarios, not all the users are experienced programmers. The proposed benchmark closes the gap between well-designed prompts and more intuitive prompts generated by normal users. Besides, collected prompts are multiple for each question and the analysis in this paper can serve as a guidance for improving the usage of Code LLMs.

**Weaknesses:**

* The way to select 48 problems can be ad-hoc and impacts the authority of evaluating Code LLM performance seriously. Especially when the problems are selected by the familiarity.
* The description and takeaways in Figure 6 should be improved. Colors in (a) are hard to understand the story behind it.
* In Section 5.4, the authors mention prompt reliability, but the way that StudentEval deals with lucky prompt is not missing. Is there any calibration or selection in StudentEval?
* The main results or findings in the analysis should be written more clearly and organized in a better way.

**Questions:**

* How does StudentEval deal with lucky prompt? Are they still included in the benchmark?
* Is StudentEval helpful to improve Code LLM performance when they are used to train the model?